# Emotional Regulation Self-Efficacy Influences Moral Decision Making: A Non-Cooperative Game Study of the New Generation of Employees

**DOI:** 10.3390/ijerph192316360

**Published:** 2022-12-06

**Authors:** Bo Liu, Taishan Yang, Wei Xie

**Affiliations:** School of Economics and Management, Southwest Petroleum University, Chengdu 610500, China

**Keywords:** new generation of employees, emotion management, communication effect, interpersonal trust, moral decision-making

## Abstract

Scholars generally believe that personality characteristics and psychological factors influence individual moral decision-making. However, few have ever discussed specific psychological factors and characteristics having such influences. Based on the self-efficacy theory and the social identity theory, this paper has proposed, from the perspective of social cognition, that emotional regulation self-efficacy influences the moral decision-making of the new generation of employees and that the mediating effect of interpersonal trust and the regulating effect of communication also play a role in the decision-making process. This study has designed a “red-blue experiment” based on the complete static information model in the non-cooperative game theory so as to conduct an experimental and qualitative analysis for the new generation of employees and to explore the characteristics of psychological process, self-efficacy, and moral decision-making of the experimental population. Through analysis of the 138 data sources collected from the experiment, the results showed that emotional self-efficacy had a significant positive effect on moral decision-making (*p* < 0.01), emotional self-efficacy had a significant positive effect on interpersonal trust (r = 0.560, *p* < 0.01), and interpersonal trust had a significant positive effect on moral decision-making (r = 0.290, *p* < 0.01). The mediating effect was 0.163. The interaction terms of emotional regulation self-efficacy and communication effect had a significant negative effect on interpersonal trust (r = −0.221, *p* < 0.01). All the hypotheses proposed in this study are supported by experimental data and reveal the psychological mechanism of moral decision-making in the new generation of employees. The study has further shown that the moral education of the new generation of employees needs to focus on improving emotional regulation self-efficacy and enhancing interpersonal trust, which provides theoretical support for the moral education methods and paths of the new generation of employees.

## 1. Introduction

The definition of the new generation of employees in China is mostly based on the year of birth, growth period, and background of the employees. The mainstream view is that the new generation of employees includes young employees who were born between 1995 and 2000, grew up with computers and the Internet, and started to work not too long ago (Hu Lili, Wang Gang, 2020) [1]. The new generation of employees is playing a pivotal role in the development of enterprises and serving as an important force for social and economic development. They have also gradually become the core groups of various organizations, exerting a profound influence on the economic, political, and social structures (Li Jin, 2020) [2]. Of course, while acknowledging the positive role of the new generation of employees in all walks of life in society, it is necessary to have an in-depth understanding of the characteristics of this group. The new generation of employees, born in accordance with the trend of the times amid the globalization of the economy, are characterized by high knowledge levels, strong learning ability, and diversified value orientation (Li Xiaofei, 2019) [3]. At the same time, the new generation of employees has poor stress resistance, low mental health, are prone to withdrawal behavior when facing huge social survival pressure, and they are characterized by a high turnover rate in the workplace (Liu Guofang, 2020) [4]. They often leave to escape the frustration of work, regardless of their responsibilities, and this phenomenon will affect the healthy development of the social economy. In professional activities, due to the complex society and the multiple needs of the main body, the new generation of employees often face different moral situations of moral decision-making (Zhao Ying, 2005) [5]. Considering the characteristics of the times and personality characteristics of the new generation of employees, it is very necessary to study their psychological processes and cognitive abilities. It is also necessary to conduct an in-depth study of the new generation of employees’ moral decision-making. Moral decision-making is the basic form of life choice. It refers to the choice of moral consciousness and moral behavior made by people in a real moral relationship according to certain moral ideals, moral principles, and moral norms. It is the choice made by human beings in the moral field (Lu Feng, 2009) [6]. It is also the individual’s final choice to weigh the pros and cons of behavior and behavioral outcomes when facing conflicts between two or more moral values or moral needs (Xu Kepeng, 2020) [7]. An unavoidable question in both theories and practices of moral decision-making is the relationship between fact and value, and moral subjects need to give their initiative and independence full play during moral decision-making through weighing and judgment (Zhao Qingwen, 2020) [8]. The new generation of employees will face different levels and situations of moral decision-making in their work. Reynolds (2010) believes that the moral decision-making of employees is the result of the environment and their own choices [9]. According to Huang Jian (2020), the moral decision-making of the new generation of employees refers to the good and evil choices or the choices of different moral values made among various possibilities or in some situations of contradictions and conflicts (Huang Jian, 2020) [10]. Essentially, it is a highly autonomous activity of human beings and is closely related to individual freedom. In moral decision-making situations, individual characteristics inevitably have an impact. As Li Zhanxing (2015) said, moral emotions have different influences on moral decision-making in different situations [11]. Negative emotions such as disgust, anger, and sadness enhance deontic judgments in moral dilemmas (Szekely, 2015) [12], while positive emotions increase utilitarian judgments (Valdesolo, 2006) [13]. Studies on patients with impaired emotional brain regions (Moll et al., 2018) [14] and those with high psychotic characteristics (Pletti, 2017) [15] also confirmed the above findings. It can be seen that emotions play an important role in the process of moral decision-making. In addition, individual efforts and beliefs (Lee, 2019) [16] will not only affect moral decision-making but also monetary stimulus (Du Xiufang, 2018) [17]. This also confirms that moral decision-making is the result of the interaction between the environment and the individual.

Numerous scholars have analyzed the influence factors of moral decision-making from different perspectives, and the subjects of choice are influenced by psychological characteristics of personality and personality inclination during moral decision-making, wherein subconsciousness, intuition, emotions, desire, imagination, etc., play mediating and regulating roles (Yang Min, 2000) [18]. From the perspective of cognitive neuroscience, the research on the relationship between psychological characteristics and moral judgment ability also shows that moral choice is influenced by psychological factors. With the improvement of the psychological quality of subjects, their moral judgment ability will be greatly enhanced, so as to make better moral choices (Liu Bo, 2018) [19,20]. Cai Xaihong and Chen Chuangsheng (2018) proposed that moral decision-making comes from the cognition of the subjects of choice, correlates with their emotions, and depends on their moral wills and socialization levels [21]. The study of Fan Ning (2022) also proved the influence of individual cognition and emotion regulation strategies on moral decision-making. This study used the time pressure paradigm and Stroop task to investigate the influence of cognitive reappraisal and expression suppression on moral decision-making under different cognitive resources [22]. Effective use of emotion regulation strategies can change the tendency of moral decision-making. Interestingly, regulatory emotional self-efficacy is an individual’s confidence in whether they can effectively regulate their emotional state, which will directly or indirectly affect various psychological functions (Tang Dongling, 2010) [23].

When the new generation of employees engages in social work, their communication and trust with others are crucial to the healthy development of enterprises. According to Zhang Wei et al. (2016), interpersonal trust refers to the dependence of individuals on others’ behaviors or commitments in the context of insufficient cognition of the environment [24]. While working in enterprises, employees with strong emotion management abilities will also be stress-tolerant. They behave more stably when communicating with others and have better dependence on others. The new generation of employees often deal with trustworthy colleagues and friends, and they cooperate more closely and communicate more fully. Colleagues who trust each other will help each other during work and make more rational choices. Employees who are more confident in controlling their own emotions and consciousness are more able to convey effective information when communicating with others, producing a better communication effect. Exploring the influencing mechanism of emotional regulation self-efficacy on moral decision-making will be conducive to understanding the psychological basis of individual behaviors and the causes of social phenomena, forming positive codes of ethics, and building a harmonious society, thus showing the social value of moral decision-making (Li Honghan, Wen Shuwen, 2017) [25]. From the literature review, existing theories have insufficient discussions on the moral situations faced by the new generation of employees. When examining individuals’ risk preferences or moral decision-making, the psychological process of decision-makers at the moment of decision-making should not be ignored (Lu Jingyi, Shang Xuesong, 2018) [26]. Although existing studies have covered multiple dimensions, including the psychological characteristics of personality and the external environment, content concerning social cognition still needs to be supplemented. Since regulatory emotional self-efficacy will affect a variety of psychological functions, and psychological functions affect moral decision-making, the author uses social cognitive theory to study the psychological impact of regulatory emotional self-efficacy on moral decision-making mechanisms. The new generation of employees is playing an increasingly important role in the economic society. As important stakeholders, employees are not only affected by the characteristics of corporate social responsibility but also affect the performance of corporate social responsibility (Jones, 2017) [27]. However, their personality characteristics often bring moral conflicts. To provide support for the innovation of moral education methods and approaches for the new generation of employees is a subject that must be paid attention to in the sustainable and healthy development of economic society, and is also the starting point and foothold of this study.

In the process of dynamic mixed multi-attribute decision-making, some scholars used the form of double-scale intuitive fuzzy number and interval value to give the intuitive fuzzy number of decision-makers in different periods and verify the proposed model (Jana, 2021) [28]. This scholar also believes that in the era of the knowledge economy, data accumulation is an important research tool (Jana, 2021) [29]. After more than half a century of rapid development, the modeling and solving of game theory for various game problems have achieved a series of milestone achievements, which have played a great role in promoting the development of economics and related fields (X. Xu, 2008) [30]. This brings inspiration and inspiration to the author. In order to understand the psychological factors of the moral decision-making of the new generation of employees and provide thinking direction and suggestions for the training of enterprise employees, combined with China’s national conditions and social development, this study uses the non-cooperative game theory to compile the “red-blue experiment”. This experiment can induce the moral choice situation through the payment matrix and capture and quantify the experimental results, so as to test and analyze whether and how the emotional regulation self-efficacy of the new generation of employees will affect the moral decision of the new generation of employees, and what roles the communication effect and interpersonal trust play in the influencing process. In the practical application of seeking the psychological factors of moral decision-making, the experimental design is natural and real, and the experimenters do not need to fill in any scale according to their subjective judgment for the data to be obtained. Effective data collection is the advantage of the “red-blue experiment”, and there is no control error. The framework of the study is shown in Figure 1.

## 2. Literature Review and Hypotheses

### 2.1. Emotional Regulation Self-Efficacy and Moral Decision-Making

The social cognitive theory requires people to be capable of self-motivation and self-improvement, and self-efficacy can influence people’s life and work in many ways (Bandura, 1988) [31]. Emotional regulation self-efficacy refers to an individual’s confidence in his or her emotional regulation ability under special situations (Connolly, 1989) [32]. Emotional regulation self-efficacy generally serves as a prerequisite for action. As the new generation of employees is generally in a poor psychological state, their moral decision-making during work tends to be selfish. Individual employees will regulate their own decision-making, thinking, and behaviors through self-efficacy (Bandura, 2006) [33]. The context of moral decision-making is no exception, and the lack of self-efficacy will directly lead to a lack of motivation (Heuven, 2006) [34]. Employees with high self-efficacy are often more willing to accept challenging work. Therefore, emotional regulation self-efficacy will act on the moral decision-making of the new generation of employees.

**Hypothesis 1.** 
*Emotional regulation self-efficacy has a positive influence on the moral decision-making of the new generation of employees.*


### 2.2. The Mediating Role of Interpersonal Trust

How does emotional regulation self-efficacy influence the moral decision-making behaviors of the new generation of employees? As shown in the existing empirical studies on emotional regulation self-efficacy, subjective well-being, depression, stress coping, etc., emotional regulation self-efficacy is closely related to psychological health. Rotter (1967) defined interpersonal trust as the generalized expectations about the reliability of commitments in interpersonal activities [35]. The interpersonal trust of the new generation of employees varies from person to person, and individuals with confident psychological characteristics are usually better at communicating with others and have higher interpersonal trust. Van (2010) proposed that interpersonal trust contributes to employees’ organizational citizenship behaviors and decision-making [36]. Girme (2015) believed that people with an avoidant attachment generally have low self-efficacy and lower trust levels [37]. Each individual has a self-regulation and self-monitoring system and will develop his or her own unique moral norms. Moral decision-making is influenced by various psychological processes. The higher the level of trust with others is, the more prominent the phenomena, such as cooperation, mutual help, and win-win, will be. Therefore, the new generation of employees with strong emotional regulation self-efficacy will have the tendency to trust others and have higher interpersonal trust and also higher moral decision-making.

**Hypothesis 2.** 
*Interpersonal trust plays a mediating role between emotional regulation self-efficacy and moral decision-making.*


### 2.3. The Mediating Role of Communication Effect

One explanation for the effect of emotional regulation self-efficacy on the new generation of employees’ interpersonal trust lies in familiarity with others, and the communication effect is the main factor deciding such familiarity. The communication effect and interpersonal trust are highly correlated (Fink, 2017) [38], and communication frequency is one of the factors influencing the communication effect. Communication frequency may be significantly correlated with emotional trust and cognitive trust (Han, 2017) [39]. As a result, in this study, the communication effect among the new generation of employees is selected as a moderating variable, which influences the relationship between emotional regulation self-efficacy and interpersonal trust, and it refers to the effect produced by the new generation of employees in the communication process. As mentioned above, the emotional regulation self-efficacy of the new generation of employees can influence several psychosocial functions, and employees with high self-efficacy will have stronger interpersonal trust. High emotional regulation self-efficacy generally means that employees are more confident when treating work and their relationship with colleagues and are willing to communicate with others and learn about others’ tasks and situations. In fact, interpersonal trust is influenced by the interaction between the communication effect and emotional regulation self-efficacy. That is to say, if there is a high communication effect among the new generation of employees, the influence of emotional regulation self-efficacy on interpersonal trust would be relatively weakened. Therefore, among the new generation of employees with a high communication effect, their emotional regulation self-efficacy can hardly influence interpersonal trust. The theoretical model of this study is shown in Figure 2.

**Hypothesis 3.** 
*The communication effect of the new generation of employees can regulate the relationship between emotional regulation self-efficacy and interpersonal trust.*


## 3. Study Methods

### 3.1. Study Methods and Samples

This study selected the experimental and interview methods and followed the study approach of model building, data analysis, and theoretical derivation. The sample subjects are the Chinese new generation of employees working in different enterprises. The study first selected a certain population and divided it into groups with 5–6 members in each one, and the members could consult with each other freely. According to the numbering, such groups were divided into multiple odd groups and even groups, and adjacent odd and even groups were asked to play games. After the end of the games, the members of the new generation of employees participating in the experiment were required to describe the process so that a return visit report could be obtained. The study has carried out 13 experiments, with each experiment consisting of 8–12 groups and each group consisting of 4–8 persons. The purpose of the experiments is to obtain the moral decision-making and state characteristics of the new generation of employees in the games. A total of 138 sets of experimental data were collected from 2015 to 2021.

### 3.2. Experimental Model and Measurements of Variables

#### 3.2.1. Experimental Model

The game theory considers the predicted and actual behaviors of individuals in games and studies individual optimization strategies. The non-cooperative game can be further divided into complete information static games, complete information dynamic games, incomplete information static games, and incomplete information dynamic games. This experiment is an improved model based on the complete information static game theory. The two sides of the game are multiple odd and even groups, and each group has two choices: red and blue. The strategic form representation is shown in Table 1 (the numbers refer to a payoff function). All groups play the game five times each. The first and second games are played without consultation between groups, the third game is played after consultation between groups, and the fourth and fifth games are played without consultation between groups but with a doubled score. The score of the odd and even groups after each game and the total score after the five games are counted, and the winning condition of the game is that the score of the group is positive after five games (Liu Bo, 2021) [40].

#### 3.2.2. Measurements of Variables

Emotional regulation self-efficacy: Bandura (2020) stated that people’s cognition of self-efficacy often changes with situations. As self-efficacy is not an overall and unchanged quantity, the use of a general self-efficacy scale will differ from the real situation, and its internal construct can hardly fit the bill. Therefore, measurements of emotional regulation self-efficacy in this study came from the qualitative analysis of the description reports of the new generation of employees. Upon carefully reading the 712 description reports, the open-coded items of emotional regulation self-efficacy were obtained, as shown in Table 2.

As can be seen, four key indicators of emotional regulation self-efficacy can be deduced from the experiment: emotional extension of collaboration with other members (items 1, 3, 4, and 6), self-disclosure during the experiment (items 2, 8, 11, and 13), control of self-consciousness and emotions (items 7, 10, and 12), and self-protection after suffering injustice (items 5 and 9). According to these indicators, adaptability scoring is performed for the emotional regulation self-efficacy of each group to obtain the measurement data.

Interpersonal trust: Schilke (2018) argued that brief contacts would influence trust decisions [41]. The more intimate the connections with others are, the more the new generation of employees’ choices will be dependent on the trust in the other group during the game. Therefore, the average score of the first and fourth games was selected as the measurement data for interpersonal trust (the score weight of the first game is 0.5, and that of the fourth game is 0.5).

Communication effect: The third game is a game with consultation between groups. Relatively, the stronger the communication effect of the new generation of employees is, the higher the corresponding score should be. Therefore, the score of the third game was selected as the measurement data of the communication effect variable.

Moral decision-making: Since the members of the new generation of employees need to think about winning the game, the more alienated they are from others, the more their moral decision-making will be dependent on their moral norms. Therefore, the average score of the second and fifth games was selected as the measurement data for interpersonal trust (the score weight of the second game is 0.5, and that of the fifth game is 0.5).

### 3.3. Statistical Processing

The study first used SPSS for brief descriptive analysis and then used AMOS to construct the mediating and regulating models for data analysis. Bootstrap was used to test the moderating effect, and the product of the independent variable of the interaction item and the moderating variable was used to test the moderating model.

## 4. Data Analysis Results

### 4.1. Descriptive Statistics

Firstly, a one-sample *T*-test and correlation analysis were carried out on the four variables to study the overall situation of the participants and explore the model prerequisites. The test results include mean (M), standard deviation (SD), and correlation coefficient (these indicators can fully describe the data characteristics), as shown in Table 3. At the end of the experiment, the average score of emotional regulation self-efficacy of the whole experimental group was 2.30, with good overall distribution. The average score for communication effect was 2.04, which is favorable compared to that of interpersonal trust (average score: 0.457) and moral decision-making (average score: 0.152). Emotional regulation self-efficacy showed a significant positive correlation with both interpersonal trust (r = 0.560, *p* < 0.01) and moral decision-making (r = 0.440, *p* < 0.01), and interpersonal trust was significantly positively correlated with moral decision-making (r = 0.290, *p* < 0.01).

### 4.2. Testing of Hypotheses

Hypothesis 1 proposed that emotional regulation self-efficacy has a direct influence on the moral decision-making of the new generation of employees. The model analysis results are shown in Figure 3. According to the results in the “red-blue experiment”, the new generation of employees with different emotional regulation self-efficacy did produce different moral decision-making, especially those with strong self-efficacy. They were very active and positive, and their moral decision-making had a tendency toward win-win cooperation. Emotional regulation self-efficacy had a significant positive influence on moral decision-making (r = 0.440, *p* < 0.01). Therefore, Hypothesis 1 is supported by actual experimental data.

Hypothesis 2 proposed that interpersonal trust plays a mediating role between emotional regulation self-efficacy and moral decision-making. After the mediation model was established, the analysis results were tested through the Bootstrap mediating effect, as shown in Figure 4. According to the fitting results of various models, emotional self-regulation had a significant positive effect on interpersonal trust (r = 0.560, *p* < 0.01), and interpersonal trust had a significant positive influence on moral decision-making (r = 0.290, *p* < 0.01). The direct, indirect, and total effects between emotional regulation self-efficacy and moral decision-making were 0.277, 0.163, and 0.440 respectively. Upon observation of the upper and lower limits of the bias-corrected confidence interval, it was found that the lower limit and upper limit of the indirect effect of emotional regulation self-efficacy on moral decision-making were 0.243 and 1.119 respectively, and the bias-corrected confidence interval did not contain 0. Therefore, emotional regulation self-efficacy has a significant mediating effect on moral decision-making through the passing of interpersonal trust. During the experiment, employees with good emotional regulation self-efficacy were found to have a stronger willingness to cooperate, and they could adjust mental attitudes timely even when they were betrayed. Trust between employees had a significant positive influence on the scores. If employees found the opposite groups to be trustworthy, their moral decision-making would tend to mutual benefit and win-win results, and the odd and even groups would choose red at the same time in the next game and get high scores. However, if the experimental groups built a psychological state of distrust, their moral decision-making would be dominated by risk aversion, and if they chose to be selfish for risk aversion (if they chose blue, they could ensure that the opposite group could not win the game), the scores would become lower and lower as the experiment goes on. The suspicious psychological state could cause the experimental groups to behave in ways that were not conducive to mutual benefits. Therefore, Hypothesis 2 is supported by actual experimental data.

Hypothesis 3 proposed that the communication effect of the new generation of employees can regulate the relationship between emotional regulation self-efficacy and interpersonal trust. After the regulation mode was established, the analysis results were tested as shown in Figure 5. The interaction item between emotional regulation self-efficacy and communication effect had a significant negative influence on interpersonal trust (r = −0.221, *p* < 0.01). For the new generation of employees with high communication effect, their emotional regulation self-efficacy would have little effect on interpersonal trust. During the experiment, when playing the game after the third communication and negotiation, the groups originally with high emotional regulation self-efficacy produced completely different moral decision-making in the subsequent games. The original emotional regulation self-efficacy of the new generation of employees had a weakened influence on interpersonal trust, which, combined with the influences of communication effect on interpersonal trust, further determined the results of moral decision-making. Before the communication with other participants, the moral decision-making made by the new generation of employees tended to be the result of emotional regulation self-efficacy. After effective communication and cognition, the final results were influenced by both factors. Therefore, Hypothesis 3 is supported by actual experimental data.

## 5. Discussion

### 5.1. Discussion of Results

Through actual testing of the new generation of employees through the “red-blue experiment”, the study found that: (1) emotional regulation self-efficacy is positively correlated with moral decision-making; (2) interpersonal trust plays a mediating role in the relationship between emotional regulation self-efficacy and moral decision-making; (3) the communication effect plays a negative role in the relationship between emotional regulation self-efficacy and interpersonal trust. Given that everyone in the new generation of employees was born in the Internet era, their psychological characteristics determine that their moral decision-making tends to be selfish. Differences in individual emotional regulation self-efficacy, level of trust with others, and the communication effect all have influences on the decision-making results.

### 5.2. Theoretical Significance

Through the interaction analysis on emotional regulation self-efficacy and moral decision-making of the new generation of employees, this study found that when conducting moral decision-making, the new generation of employees is influenced by a variety of factors, including self-efficacy, interpersonal trust, and the communication effect, which has revealed the influence mechanism of their internal psychological process on moral decision-making. Studies of moral decision-making are not restricted to individuals but can also be interpreted from the perspective of psychological cognition to understand the interest appeal and role-play of the new generation of employees during moral decision-making from the psychological and behavioral development level and explore the changing rules and formation mechanism. To guide the moral quality of the new generation of employees to the right road, we need to increase their self-efficacy and trust in organizations and colleagues.

### 5.3. Practical Significance

This study has understood the operating mechanism of the new generation of employees during moral decision-making. For the moral education of the new generation of employees, enterprises shall not only cultivate their behavior quality but also strengthen their psychological health education. They need to establish a good trusting atmosphere, improve the emotional regulation self-efficacy of the new generation of employees, and provide assessments and suggestions for their physical and mental health. Therefore, the study has provided a reference for employees’ moral quality education in enterprises.

### 5.4. Limitations of the Study

Three limitations of this study should be mentioned: (1) The samples were collected in large cities only, so the sample diversity is insufficient. Additionally, only 138 data points were used, which is insufficient in amount and will limit the credibility of the samples to a certain extent. Subsequent studies can expand the size of samples to conduct research in a wider range. (2) Since the experimental framework of this study adopts the complete information game model in the non-cooperative game theory only, the game model needs to be supplemented. (3) This experiment did not use monetary incentives and punishments to obtain conditional results, so the experimental content needs to be expanded.

## 6. Conclusions

As the new generation of employees gradually becomes the main force in the development of all walks of life, they are exposed to significant conflicts in different situations of moral decision-making, which not only hinders the development of individuals and organizations but also affects the establishment of good moral norms in the society. In order to reveal the psychological mechanism behind the moral decision-making of the new generation of employees and explore the influencing mechanism of their moral decision-making, the “red-blue experiment” was designed to capture the effect of psychological factors under the situation of moral choice. Based on the experimental data of the “red and blue experiment”, a moderated mediation effect model was constructed to empirically test the path of emotional regulation self-efficacy under the regulation of communication effectiveness to affect moral decision-making through interpersonal trust. The indirect effect of emotional regulation self-efficacy on moral decision-making was 0.163, and the total effect was 0.440. The mediating effect of interpersonal trust between emotional regulation self-efficacy and moral decision-making was significant. Employees with a trusting attitude are more likely to cooperate in their moral decisions, while once they have a distrusting attitude, they are more likely to choose to avoid risks and make conservative choices. The interaction term of emotional regulation self-efficacy and communication effect has a significant negative impact on interpersonal trust (*p* < 0.01), which means that emotional regulation self-efficacy plays a weak role in interpersonal trust, and often the new generation of employees has a high communication effect.

The results of the comprehensive analysis show that individual emotional regulation self-efficacy differences, interpersonal trust level, and the communication effect can affect the outcome of moral decision-making. From the internal psychological mechanism, when the new generation of employees faces an unfamiliar environment and group, their moral decisions are relatively selfish and conservative. Under the influence of various factors, such as the change of mental state and game victory, the new generation of employees with different emotional regulation self-efficacy and interpersonal trust will have their own choices. Empirical research on moral decision-making theory lacks comprehensive consideration of cognition and emotion (Chen, 2020) [42]. However, the innovation of this research model and method provides a new perspective for finding the psychological factors of moral decision-making of the new generation of employees and exploring the psychological mechanism of moral decision-making of the new generation of employees and expands the related research in the field of moral decision-making. It also provides a new idea for enterprise managers to pay attention to the moral training of the new generation of employees, which should develop their social cognitive ability and give a better atmosphere and trusting environment.

## Figures and Tables

**Figure 1 ijerph-19-16360-f001:**
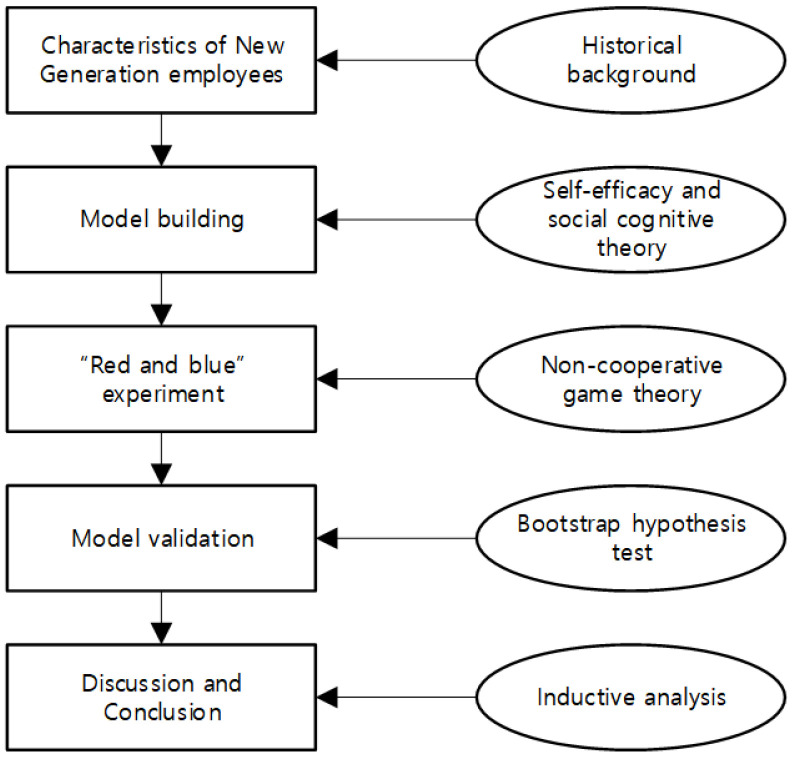
Framework of the study.

**Figure 2 ijerph-19-16360-f002:**
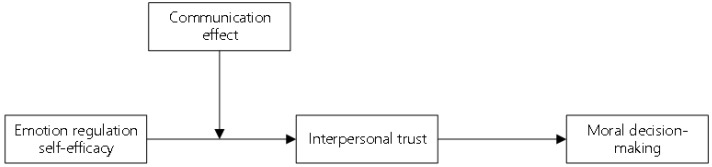
Theoretical model.

**Figure 3 ijerph-19-16360-f003:**
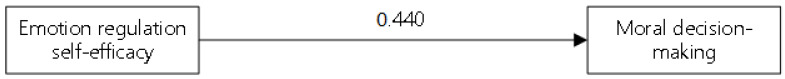
Influence of emotional regulation self-efficacy on moral decision-making.

**Figure 4 ijerph-19-16360-f004:**
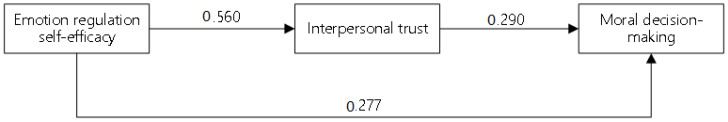
The mediating role of interpersonal trust between emotional regulation self-efficacy and moral decision-making.

**Figure 5 ijerph-19-16360-f005:**
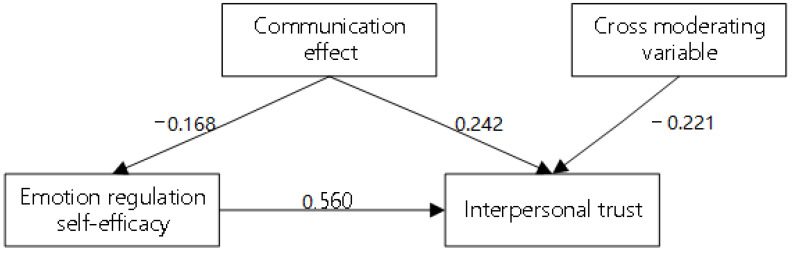
The moderating role of moral consciousness between emotional regulation self-efficacy and interpersonal trust.

**Table 1 ijerph-19-16360-t001:** Game matrix of the “red-blue” experiment.

Payoff Matrix		Even Group	
		Red	Blue
Odd group	Red	+3, +3	−6, +6
	Blue	+6, −6	−3, −3

**Table 2 ijerph-19-16360-t002:** Open-coded items of emotional regulation self-efficacy.

Coded Item	Content
1	Trust can hardly be built up again once broken
2	Win-win results and honesty are the best choices
3	We still trust each other resolutely even after seeing other groups’ betrayal
4	Since betrayal could bring terrible results, after a lot of struggle, we honored the commitments
5	Angry at being betrayed, we chose blue
6	We both thought the other would be selfish
7	We chose blue driven by revenge, while the other side chose red because of guilt
8	Having a longer-term view, balancing benefits, and daring to take risks and the consequences
9	Expressing remorse and ultimately tending to limit the losses
10	Consciously choosing to avoid risk and cutting losses
11	Either giving it a gamble or taking things as they are
12	Choosing to honor the commitments
13	Trust is risky. Principles are immutable

**Table 3 ijerph-19-16360-t003:** Mean, standard deviation, and correlation coefficient (*p* * < 0.05, *p* ** < 0.01).

Variable	M	SD	Emotional Regulation Self-Efficacy	Communication Effect	Interpersonal Trust	Moral Decision-Making
Emotional regulation Self-efficacy	2.30	1.168				
Communication effect	2.04	3.221	−0.168 *			
Interpersonal trust	0.457	4.471	0.560 **	−0.242 **		
Moral decision-making	0.152	4.7961	0.440 **	0.018	0.290 **	

## Data Availability

The data presented in this study is available on request from the corresponding author. The data is not publicly available due to data privacy.

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
