# Peer review of "Emotional Regulation Self-Efficacy Influences Moral Decision Making: A Non-Cooperative Game Study of the New Generation of Employees"

_ijerph, 2022, doi:10.3390/ijerph192316360_

Round 1

Reviewer 1 Report

In this  paper, the authors have proposed from the perspective of social cognition that emotional regulation self-efficacy influences the moral decision-making of the new generation of employees and that the mediating effect of interpersonal trust and the regulating effect of communication also play a role in the decision-making process. To test the viewpoint hypothesis, this study has designed a “red-blue experiment” based on the complete static information model in the non-cooperative game theory so as to conduct an experimental and qualitative analysis for the new generation of employees and to explore the characteristics of psychological process, self-efficacy, and moral decision-making of the experimental population. Through analysis of the 138 data sources collected from the experiment, the study has found that the moral decision-making of the new generation of employees is influenced by self-efficacy, interpersonal trust, communication quality, and a variety of other factors, which reveals the psychological mechanism of the new generation of employees during the moral decision-making. The paper is good and interesting but some major improvement is required as follows:

1)    I suggest the authors to improve the introduction section. Authors should better highlight the objective of their work and to what extent it contributes to close a gap in the existing literature and/or practice. What is the innovative value of the contribution proposed by the authors?

2)    The authors must clearly discuss the significance of the research problem in the first section.

3)    You should provide more recent references published in last two-three years. Remove references published before 2017. Also, remove lumped references, all references should be discussed in the manuscript. Some recent interesting references from the MCDM field are missing
 I suggest authors to read and discuss the below listed references:

A dynamical hybrid method to design decision making process based on GRA approach for multiple attributes problem, Engineering Applications of Artificial Intelligence 100, 104203, 2021;

Intuitionistic fuzzy dombi hybrid decision-making method and their applications to enterprise financial performance evaluation, Mathematical Problems in Engineering 2021; 

4)    How should we know about the quality of these solutions? Could you compare these results with  more existing approaches? The novelty must be discussed.

5)    Validation and robustness section is missing. Conclusion section should be updated by adding the advantages and contribution of the work

Author Response

Dear  experts,
   Thank you very much for your valuable advice. All your suggestions are very important and have important guiding significance for my thesis writing and scientific research work. According to your suggestion, the paper has been modified as follows.We have made detailed modifications to your suggestions, and we have sent the reply to each of them in the form of PDF attachmen.

Reviewer 2 Report

In my opinion, the results are novel, and interesting. The paper is suitable for publication in IJERPH, however the following revision is needed before its final version:

1. The Abstract section needs more work. It should be outcomes-oriented. Please rewrite it. 

2. The motivation for this work and the background of available techniques are not clearly described in the introduction. In principle, the selected approach and theory are not properly described, though some improvements are necessary. Please give a short explanation of how the references in the introduction relate to the approach you are suggesting. 

3. What characteristics of the adopted methods make them a superior fit for the problem considered in this manuscript.

4. Explain the merits and demerits of the methods used in practical applications. 

5. The conclusion can be strongly revised. This reviewer strongly suggests improving the flow of the conclusion section. Start with a brief explanation of the paper's goal (like the abstract), but make sure that the conclusion is different from the abstract. Provide the main findings/claims. Explain the numerical findings of the simulations. Clearly explain what the significant findings are and why your paper is really important.

Author Response

Dear  experts,
   Thank you very much for your valuable advice. All your suggestions are very important and have important guiding significance for my thesis writing and scientific research work.According to your suggestion, the paper has been modified as follows.We have made detailed modifications to your suggestions, and we have sent the reply to each of them in the form of PDF attachmen.
